# Multidrug-resistant and carbapenemase-producing critical gram-negative bacteria isolated from the intensive care unit environment in Amhara region, Ethiopia

**Mizan Kindu**[1]*, **Feleke Moges**[1], **Degu Ashagrie**[2], **Zemene Tigabu**[3], **Baye Gelaw**[1]

**1** Department of Medical Microbiology, University of Gondar, Gondar, Ethiopia, **2** Medical Microbiology Laboratory, Felege Hiwot Comprehensive Specialized Hospital, Bahir Dar, Ethiopia, **3** Department of Pediatrics and Child health, University of Gondar, Gondar, Ethiopia

* mizankindu00@gmail.com

## Abstract

### Background

Intensive care units are units where healthcare-associated infections (HAIs) are common and antimicrobial resistance rates are increasing. Microbial contamination in hospital environment plays an important role in the development of HAIs. Intervention-based improvements in infection prevention and control at national and facility level are critical for the containment of antimicrobial resistance and prevention of HAIs.

### Objectives

This study aimed to determine the distribution of multidrug-resistant and carbapenemase-producing critical gram negative bacteria (*Klebsiella pneumoniae*, *Escherichia coli*, *Pseudomonas aeruginosa* and *Acinetobacter* species) and their antibiotic resistance in intensive care unit environmental surfaces at the University of Gondar and Felege Hiwot Comprehensive Specialized Hospitals.

### Methods

This was multicenter hospital-based cross sectional study. Environmental samples were swabbed from all intensive care units using a normal saline moistened-sterile cotton tip stick. Bacteria culturing and antibiotic susceptibility testing were performed following standard microbiological techniques. Selected meropenem-resistant isolates were phenotypically assessed for carbapenemase production using modified and simplified carbapenem inactivation methods.

### Results

From a total of 384 environmental samples analyzed, 126 (32.8%) showed growth and 162 isolates were identified. *K. pneumoniae* (79/162, 48.8%) was the commonest isolate followed by *Acinetobacter* species (51/162, 31.5%), *E. coli* (19/162, 11.7%) and *P. aeruginosa*

**Funding:** The author(s) received no specific funding for this work.

**Competing interests:** The authors have declared that no competing interests exist.

(13/162, 8.0%). Multidrug-resistant and carbapenemase-producing isolates were detected on most hospital environment surface types, especially from the baby bed sets and incubators. The most common multidrug-resistant and principal carbapenemase producer was *K. pneumoniae*, with rates of 71(89.9%) and 24(85.7%), respectively.

## Conclusion

This study revealed the distribution of multidrug-resistant and carbapenemase-producing critical gram negative bacteria in the environment of intensive care unit. Higher detection rate of multidrug-resistant and carbapenemase-producing *K. pneumoniae* on most environmental surfaces calls for urgent control action and further attention.

## Introduction

Intensive care units (ICUs) are units where healthcare-associated infections (HAIs) common and antimicrobial resistance (AMR) rates are increasing [1]. In recent years HAIs in ICUs have been caused by gram-negative bacteria (GNB), mostly by *K. pneumoniae*, *E. coli*, *P. aeruginosa and Acinetobacter* species (critical-GNB), and antibiotic resistance in these pathogens has implications beyond the immediate issues of morbidity and mortality [1–3].

Overuse of antimicrobial agents and problems with infection control practices have led to the development of multidrug-resistant (MDR) GNB infections. Carbapenems are often used for the treatment of patients with MDR infections in the ICU. However, carbapenemase production by GNB worsens the treatment options. Moreover, the majority of carbapenemase-encoding genes are carried by mobile elements, such as plasmids, which might ease the horizontal transmission of resistance across other GNB in the same setting [4–6].

Microbial contamination of the hospital environment plays an important role in the development of HAI [7]. It is estimated that more than 25% of the cases of HAIs are triggered by microorganisms present in the environment, for example high touch surfaces, which leads to a greater risk of the transmission of infections in healthcare services [8]. A study conducted in Algeria showed that strains isolated from the hospital environment had a clonal relationship with those isolated from clinical samples [9].

Consequently, intervention-based improvements in infection prevention and control (IPC) at national and facility levels are critical for the containment of antimicrobial resistance and prevention of HAIs, including outbreaks of highly transmissible diseases, through high-quality care in the context of universal health coverage [10]. Understanding the contamination of ICU environments by MDR and carbapenemase-producing critical GNB is essential for evidence-based prevention strategies for HAI control. Therefore, this study aimed to determine the distribution of MDR and carbapenemase-producing critical GNB and their antibiotic resistance on intensive care unit environmental surfaces at the University of Gondar and Felege Hiwot comprehensive specialized hospitals.

## Materials and methods

### Study design and area

Multicenter hospital-based cross-sectional study was conducted at the University of Gondar Comprehensive Specialized Hospital (UoGCSH) and Felege Hiwot Comprehensive Specialized Hospital (FHCSH) between November 2021 and December 2022. The intensive care units

of these comprehensive specialized hospitals are the big and serve as a referral hospitals for critical patients from other parts of the Amhara region.

The UoGCSH has neonatal, pediatric, medical, and surgical ICUs. At the time of data collection, the surgical and medical ICUs were located in single room with eight beds owing to reinovation. The pediatric ICU has one room with six beds. The neonatal ICU has 5 rooms with more than 40 baby beds and incubators. On the other hand FHCSH has two ICUs one for neonates and one for other than the neonates. The neonatal ICU has 9 rooms, with 20 and 18 beds for pre-term and full-term neonates, respectively. The ICU for the critical care of pediatric, adult, and surgical patients has two rooms with 12 beds.

## Sample size and sampling techniques

Using a single population proportion formula, 384 samples from hospital environment were included for the study. Using simple random sampling technique, samples from hospital environment were collected from high-touch surfaces of baby incubators, baby bed sets, bed sheets, bed mattresses, bedside rail surfaces, examination tables, overbed tables, stethoscopes and sphygmomanometers. High-touch surfaces are those with frequent contact with the hands, which pose the greatest risk of transmission of microorganisms [7]. Samples from hospital environment were also collected from oxygen sets, machines, chart tables, IV sets, indoor knobs and sinks. The standardized swab surface area (not greater than 10 cm$^2$) for each selected item was swabbed using a normal saline-moistened sterile cotton tip stick [11]. All samples were sent to the microbiology laboratory immediately after collection.

## Bacterial identification and antibiotic susceptibility testing

After delivery of the samples to the laboratory, each swab was inoculated on to MacConkey agar and incubated at 37 ˚C for 18–24 hr. Preliminary identification of bacteria was done based on colony characteristics of grown isolates on media. Morphologically single distinct colonies were isolated and purified by subculturing into fresh blood agar plate (BAP) medium to obtain pure culture isolates [12]. *K. pneumoniae* and *E. coli* isolates were identified using different biochemical tests such as triple sugar iron agar, indole, motility, urease production, hydrogen sulfide production, citrate utilization and lysine decarboxylase tests. For the identification of *P. aeruginosa* and *Acinetobacter* species, in addition to morphological characteristics of the isolates and the mentioned biochemical tests, further catalase and oxidase strip tests were done.

**Antibiotic susceptibility testing.** The Kirby–Bauer disk diffusion method was used with Mueller–Hinton agar to determine the antibiotic susceptibility patterns of the isolates, and CLSI M100 was used to interpret the results [13].

The following antibiotic discs were used: penicillins (ampicillin 10μg), aminoglycosides (amikacin 30μg and gentamicin 10μg), penicillins +inhibitors (amoxycillin/ clavulanic acid 20/ 10μg), phenicols (chloramphenico l30μg), third and fourth generation cephalosporins (cefepime 30μg, ceftazidime 30μg and ceftriaxone 30μg), tetracyclines (tetracycline 30μg), fluoroquinolones (ciprofloxacin 5μg), antipseudomonal penicillins + inhibitors (piperacillin/ tazobactam 100/10 μg), folate pathway inhibitors (trimethoprim-sulfamethoxazole 1.25/ 23.75μg), first and second generation cephalosporins (cefazolin 30μg and cefuroxime 30μg) and carbapenems (meropenem 10μg). Isolates resistant to one or more antibiotic types in three or more antibiotic classes were considered multidrug-resistant [14].

## Phenotypic detection of carbapenemase

Carbapenemase production in *K. pneumoniae*, *E. coli* and *P. aeruginosa* was detected using a modified carbapenem inactivation method (mCIM). A simplified carbapenem inactivation method (sCIM) was used for *Acinetobacter* species.

**Modified carbapenem inactivation method.** The mCIM prepared by emulsifying 1μL loop-full of *K. pneumoniae* and *E. coli* or 10 μL loop of *P. aeruginosa* from blood agar plates was emulsified in 2 mL trypticase soy broth (TSB). A meropenem (10μg) disk was immersed in the suspension and incubated for a minimum of 4 hr at 37 ˚C. A 0.5 McFarl and suspension of *E. coli* ATCC 25922 was inoculated onto Mueller–Hinton agar (MHA) plates. The meropenem disk was then removed from the TSB and placed on an MHA plate inoculated with *E. coli* ATCC 25922 indicator strains. All the plates were incubated at 37 ˚C for 18–24 hr. An inhibition zone with a diameter of 6–15 mm or colonies within a 16–18 mm zone was considered a positive result, and a zone of inhibition of ≥19 mm was considered a negative result [13].

**Simplified carbapenem inactivation method.** For sCIM, 0.5 McFarl and standard suspension of *E. coli* ATCC 25922 was diluted 1:10 in saline and inoculated onto the MHA plate following the routine disk diffusion procedure [15].

Then, 1–3 colonies of *Acinetobacter* species grown overnight on blood agar were smeared onto one side of an imipenem disk (10μg); immediately afterward, and the side of the disk containing the bacteria was immediately placed on the MHA plate previously inoculated with *E. coli* ATCC 25922. An imipenem disk placed on an MHA plate was used as the control [15]. The plates were then incubated at 37 ˚C for 16–18 hr. Bacterial strains that produce carbapenemase hydrolyze imipenem; hence, susceptible indicator strains grow unrestrictedly. If the zone of inhibition around the disk had a diameter of 6–20 mm or the satellite growth of colonies of *E. coli* ATCC 25922 around the disk had a zone diameter of ≤22 mm, the result was considered carbapenemase-positive; a zone of inhibition of ≥26 mm was considered to be a negative result; a zone of inhibition of 23–25 mm was considered to be a carbapenemase indeterminate result [15].

## Data processing, analysis and interpretation

Data entry and analysis was performed using statistical package for social sciences (SPSS) Version 26 software. Descriptive statistics were calculated using frequencies and cross-tabulations.

## Ethical approval

The study was reviewed by Institutional Review Board of the University of Gondar for ethical approval. Following the review by the Institutional Review Board and recommendation, the Research and Community Service Vice President Office awarded ethical clearance for the investigators (Registration number: O/VIP/RCS/05/1773/2019). Although the study did not directly enrolled patients, verbal consent was requested from ICU medical professionals for sample collection from the environmental surface of the ICUs.

## Results

A total of 384 hospital environment samples from neonatal 246/384 (64.1%) and other intensive care units 138/384 (35.9%) were analyzed. Of the three-hundred and eighty four, 126 (32.8%) samples were tested positive for bacterial isolates. From 126 (32.8%) culture positive samples, 162 (42.2%) critical GNB isolates were identified and 36 (9.4%) samples showed mixed growth. Most of the hospital environment surfaces and equipment types were contaminated with critical GNB isolates (Table 4); however, none of the isolates were detected on

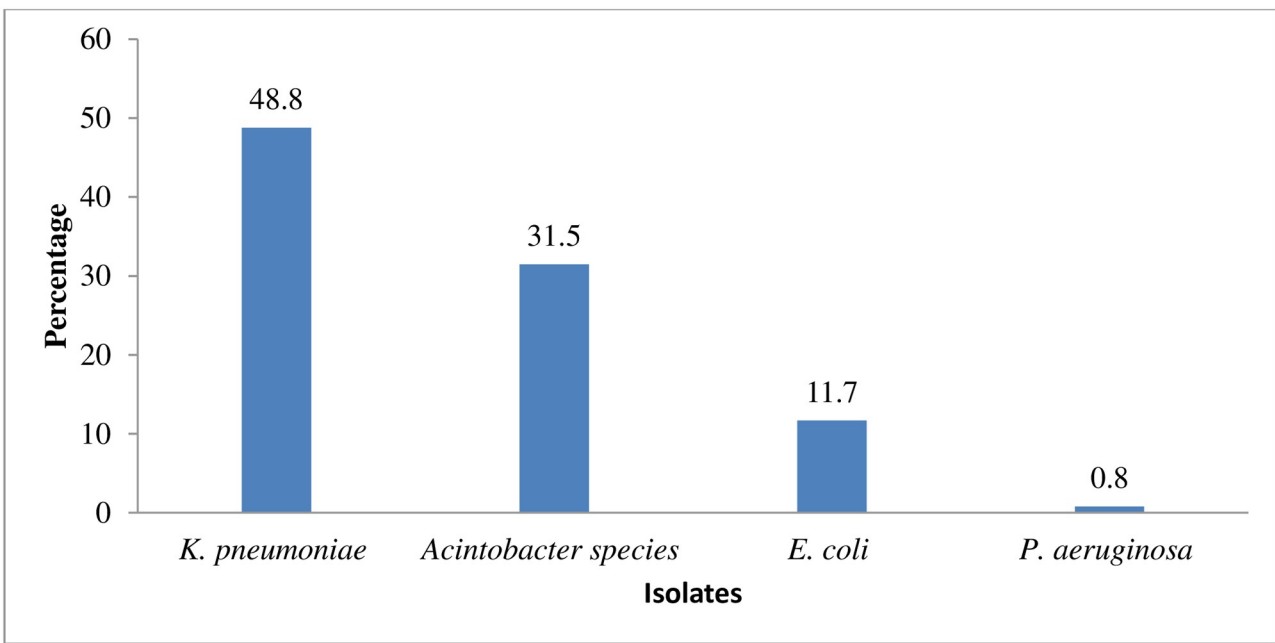

**Fig 1. Critical GNB isolated from environmental samples of ICUs at Amhara region, Ethiopia, 2021 to 2022.**

machines (n = 15), indoor knobs (n = 8), stethoscopes (n = 8) and sphygmomanometers (n = 8). The predominant isolates were *K. pneumoniae* 79/162 (48.8%) followed by *Acinetobacter* species 51/162 (31.5%), *E. coli* 19/162 (11.7%), and *P. aeruginosa* 13/162 (8.0%) (Fig 1).

### Antibiotic resistance patterns of environmental isolates

Fifteen antibiotics were used to assess resistance profiles. *E. coli* had showed 100% resistant to ampicillin. Isolates also showed higher resistance to tetracycline, cotrimoxazole, amoxicillin/clavulanic-acid, and cephalosporins, with rates ranging between 67% and 91.8%. In contrast, amikacin was the most effective antibiotic with sensitivity rate of 92.6%. Resistance to meropenem was highest in *P. aeruginosa* followed by *Acinetobacter* species, *K. pneumoniae* and *E. coli* (Table 1).

### Multidrug-resistance patterns of environmental isolates

According to Magiorakos *et al.* [14], the 15 tested antibiotics were grouped into 11 categories. Based on this category, 128(79%) isolates were classified as MDR. The MDR rate was highest for *K. pneumoniae* followed by *E. coli*, *Acinetobacter* species, and *P. aeruginosa* (Table 2). Sixty-one critical GNB isolates exhibited resistance to meropenem (28 *K. pneumoniae*, 21 *Acinetobacter* species, eight *P. auerginosa* and four *E. coli*) (Table 1). Of the MDR isolates, 33(54.1%) were carbapenemase producers (Table 2). Moreover, carbapenemase production was highest in *K. pneumoniae* followed by *P. aeruginosa* and *E. coli* and *Acinetobacter* species.

### Distribution of environmental isolates in the ICUs of the two compressive specialized hospitals

More than half of samples from hospital environment were collected at the University of Gondar Comprehensive Specialized Hospital. The culture positivity of hospital environment

**Table 1. Antibiotic resistance pattern of critical GNB in environmental samples from the ICUs of the two comprehensive specialized hospitals of Amhara region, Ethiopia, 2021 to 2022.**

| Antibiotic classes | Antibiotics types | K. pneumoniae = 79 | | E. coli = 19 | | Acinetobacter spp = 51 | | P. aeruginosa = 13 | | |
|---|---|---|---|---|---|---|---|---|---|---|
| | | S = N(%) | R = N(%) | S = N(%) | R = N(%) | S = N(%) | R = N(%) | S = N(%) | R = N(%) | Total = N(%) |
| Penicillins | AMP | - | - | 0(0%) | 19(100) | - | - | - | - | 19(100) |
| Aminoglycosides | GEN | 25(31.6) | 54(68.4) | 11(57.9) | 8(42.1) | 25(49) | 26(51) | 9(69.2) | 4(30.8) | 92(56.8) |
| | AMK | 72(91.1) | 7(8.9) | 18(94.7) | 1(5.3) | 48(94.1) | 3(5.8) | 12(92.3) | 1(7.7) | 12(7.4) |
| Beta-lactam combination group | AMC | 11(13.9) | 65(86.1) | 1(5.3) | 18(94.7) | - | - | - | - | 86(87.8) |
| Antipseudomonal-penicillins + inhibitors | TZP | 43(54.4) | 36(45.6) | 6(31.6) | 13(68.4) | 27(52.9) | 24(47.1) | 7(53.8) | 6(46.2) | 79(48.8) |
| 1st and 2nd generation cephalosporins | CZ | 7(8.9) | 72(91.1) | 1(5.3) | 18(94.7) | - | - | - | - | 90(91.8) |
| | CXM | 6(7.5) | 73(92.4) | 3(15.8) | 16(84.2) | - | - | - | - | 89(90.8) |
| 3rd and 4th generation cephalosporins | CRO | 9(11.4) | 70(88.6) | 4(21.1) | 15(78.9) | 3(5.9) | 48(94.1) | - | - | 133(82.1) |
| | CAZ | 10(12.7) | 69(87.3) | 4(21.1) | 15(78.9) | 6(11.8) | 45(88.2) | 4(30.8) | 9(69.2) | 138(85.2) |
| | CEP | 13(16.5) | 66(83.5) | 4(21.1) | 15(78.9) | 12(23.5) | 39(76.5) | 4(30.5) | 9(69.2) | 129(79.3) |
| Carbapenem | MER | 48(64.6) | 28(35.4) | 15(78.9) | 4(21.1) | 30(58.8) | 21(41.2) | 5(38.5) | 8(61.5) | 61(37.7) |
| Fluoroquinolones | CIP | 33(41.8) | 46(58.2) | 5(26.3) | 14(73.7) | 24(47.1) | 27(52.9) | 7(53.8) | 6(46.2) | 93(57.4) |
| Phenicols | CHL | 49(62) | 30(38) | 14(73.7) | 5(26.3) | - | - | - | - | 35(35.7) |
| Folate pathway inhibitors | SXT | 17(21.5) | 62(78.5) | 3(15.8) | 16(84.2) | 14(27.5) | 37(72.5) | - | - | 115(77.2) |
| Tetracyclines | TE | 28(35.4) | 51(64.6) | 4(21.1) | 15(78.9) | 17(33.3) | 34(66.7) | - | - | 100(67) |

**Key**: S = Sensitive, R = Resistance, AMP = Ampicillin; AMK = Amikacin; AMC = Amoxicillin/clavulanic acid; TZP = Piperacillin/tazobactam;

SXT = Sulphamethoxazole-trimethoprim(cotrimoxazole); TE = Tetracycline; CIP = Ciprofloxacin; CHL = Chloramphenicol; GEN = Gentamycin; CRO = Ceftriaxone; CAZ = Ceftazidime; CXM = cefuroxime; CZ = Cefazolin; CEP = Cefepime; MER = Meropenem.

samples from the ICUs was 33.8% for UoGCSH (71/210) and 31.6% for FHCSH (55/174). Moreover, the magnitudes of MDR critical GNB were 75/95(78.9%) and 53/67(80.5%) at UoGCSH and FHSCH, respectively. The contamination rate of neonatal intensive care unit at UoGCSH by MDR and CP-critical GNB was 80.8% and 60.3%, respectively whereas 70.6% and 0% for other ICUs. On the other hand, at the FHCSH, the overall contamination rate by MDR and CP-critical GNB of pediatric, adult, and surgical critical ICUs was 83.3% and 56.3%, respectively whereas 74.2% and 45.5% for the NICUs. K. pneumoniae was the most commonly detected MDR and CP isolate (Table 3).

**Table 2. Multidrug-resistance pattern and carbapenemase production of critical GNB in environmental samples from the ICUs of the two comprehensive specialized hospitals of Amhara region, Ethiopia, 2021 to 2022.**

| Isolates | R0 | R1 | R2 | R3 | R4 | R5 | R6 | R7 | R8 | R9 | R10 | R11 | MDR≥3 | CP(N(%)) | |
|---|---|---|---|---|---|---|---|---|---|---|---|---|---|---|---|
| | | | | | | | | | | | | | | -ve | +ve |
| K. pneumoniae = 79 | 6 | 0 | 2 | 5 | 4 | 6 | 9 | 14 | 10 | 10 | 13 | - | 71(89.9) | 4(14.3) | 24(85.7) |
| E. coli = 19 | 0 | 1 | 1 | 0 | 1 | 1 | 1 | 0 | 5 | 5 | 3 | 1 | 17(89.5) | 2(50) | 2(50) |
| Acinetobacter spp = 51 | 3 | 5 | 11 | 2 | 0 | 3 | 12 | 15 | - | - | - | - | 32(62.7) | 19(85.7) | 3(14.3) |
| P. aeruginosa = 13 | 3 | 1 | 1 | 2 | 2 | 4 | - | - | - | - | - | - | 8(61.5) | 4(50) | 4(50) |
| Total = 162 | 12 | 7 | 15 | 9 | 7 | 14 | 22 | 29 | 15 | 15 | 16 | 1 | 128(79.0) | 28(45.9) | 33(54.1) |

Note; R0: sensitive for all classes of antibiotics, R1: resistant for one class of antibiotics, R2: resistant for two classes of antibiotics, R3: resistant for three classes of antibiotics, etc., MDR = multidrug-resistant, CP = carbapenemase-producing

**Table 3. Distribution of MDR and carbapenemase-producing isolates from environmental samples of the ICUs of the two comprehensive specialized hospitals of Amhara region, Ethiopia 2021 to 2022.**

| Isolates | | UoGCSH | | FHCSH | | Total (N(%)) |
|---|---|---|---|---|---|---|
| | | NICU | ICUs | NICU | ICUs | |
| *K. pneumoniae* = 79 | MDR | 39* | 3 | 13 | 16* | 71(89.9) |
| | CP | 14 | - | 4 | 6 | 24(31.6) |
| *E. coli* = 19 | MDR | 12 | - | 3 | 2 | 17(89.5) |
| | CP | 1 | - | 1 | 0 | 2(10.5) |
| *Acinetobacter* spp = 51 | MDR | 9* | 9 | 4 | 10* | 32(62.7) |
| | CP | 1 | - | - | 2 | 3(7.3) |
| *P. aeruginosa* = 13 | MDR | 3 | - | 3 | 2 | 8(61.5) |
| | CP | 3 | - | - | 1 | 4(30.8) |
| Total (N(%)) | MDR | 63/78(80.8) | 12/17(70.6) | 23/31(74.2) | 30/36(83.3) | 128/162(79.0) |
| | CP | 19/31(61.3) | 0/3(0) | 5/11 (45.5) | 9/16(56.3) | 33/61(54.1) |

* = mixed growth; MDR = multidrug-resistant, CP = carbapenemase-producing

## Distribution of MDR and carbapenemase-producing isolates in ICUs environment

All culture-positive hospital environment surfaces were contaminated with at least one MDR critical GNB. Except for the oxygen concentrator, all culture-positive items were contaminated with CP critical GNB. The MDR and CP *K. pneumoniae* isolates were identified in most samples from hospital environment, and the number was higher in the baby incubators and baby bed sets (Table 4). Carbapenemase-producing *K. pneumoniae*, CP *E. coli*, and CP *P. aeruginosa* isolates were detected in sinks. Likewise, bed rail surfaces, baby bed sets, and overbed tables were contaminated with two different CP critical GNB.

**Table 4. Distribution of MDR and carbapenemase-producing isolates in ICUs environment surfaces of the two comprehensive specialized hospitals of Amhara region, Ethiopia, 2021 to 2022.**

| Sample source | Isolates | | | | | | | |
|---|---|---|---|---|---|---|---|---|
| | *K. pneumoniae* = 79 | | *E. coli* = 19 | | *Acinetobacter* spp = 51 | | *P. aeruginosa* = 13 | |
| | MDR | CP | MDR | CP | MDR | CP | MDR | CP |
| Baby incubator = 43 | 20 | 8 | 4 | - | 4 | - | - | - |
| Baby bed set = 73 | 18 | 5 | 6 | - | 6 | 1 | - | - |
| Bed rail surface = 30 | 7 | 1 | 3 | - | 5 | 2 | 1 | - |
| Over bed table = 29 | 7 | 3 | - | - | 5 | - | 1 | 1 |
| Bed sheet = 30 | 5 | - | - | - | 7 | - | 1 | 1 |
| Baby examination table = 27 | 3 | 2 | - | - | 1 | - | - | - |
| Mattresses = 20 | 4 | - | 2 | 1 | 1 | - | - | - |
| Oxygen concentrator = 21 | - | - | - | - | 1 | - | - | - |
| IV set = 25 | 3 | 2 | 1 | - | 1 | - | - | - |
| Sink = 27 | 3 | 3 | 1 | 1 | 1 | - | 4 | 1 |
| Chart table = 20 | 1 | - | - | - | - | - | 1 | 1 |
| Total | 71(89.9) | 24(85.7) | 17(89.5) | 2(50) | 32(62.7) | 3(14.3) | 8(61.5) | 4(50) |

MDR = multidrug-resistant, CP = carbapenemase-producing

### Antibiotic resistance pattern of MDR and carbapenemase-producing isolates

*K. pneumoniae* was the principal MDR and carbapenemase producer in ICUs environment. The MDR critical isolates exhibited higher rates of resistance to ampicillin, amoxicillin/clavulanic-acid, and all cephalosporins, with rates ranging from 88.2 to 100%. Similarly, all CP *K. pneumoniae*, CP *E. coli*, CP *P. aeruginosa* and CP *Acinetobacter* species isolates were 100% resistant to these antibiotics. The resistance rate of CP *K. pneumoniae* isolates were 25% to amikacin, 62.5% to ciprofloxacin, 70.8% to chloramphenicol, 75% to piperacillin-tazobactam, and 87.5% to gentamicin, sulfamethoxazole-trimethoprim, and tetracycline. All CP *E. coli* isolates showed 100% resistance to all antibiotics tested, except amikacin (50%) and chloramphenicol (0%). CP *Acinetobacter* species exhibited resistance rates of 33.3% for amikacin and 66.6% for ciprofloxacin. CP *P. aeruginosa* isolates showed 25%, 50%, 75% resistance to amikacin, gentamicin, ciprofloxacin and piperacillin-tazobactam, respectively (Table 5).

**Table 5. Antibiotic resistance pattern of MDR and carbapenemase-producing isolates in ICUs environment of Amhara region, Ethiopia, 2021 to 2022.**

| Antibiotics | | Multidrug-resistant (MDR) | | | | Carbapenemase-Producing(CP) | | | | |
|---|---|---|---|---|---|---|---|---|---|---|
| | | *K. pneumoniae* = 71 | *E. coli* = 17 | *Acinetobacter* spp = 32 | *P. aeruginosa* = 8 | *K. pneumoniae* = 24 | *E. coli* = 2 | *Acinetobacte* rspp = 3 | *P. aeruginosa* = 4 | |
| AMP | S | - | 0 | - | - | - | 0 | - | - | |
| | R | - | 17(100) | - | - | - | 2(100) | - | - | |
| GEN | S | 17(23.9) | 9(52.9) | 6(18.7) | 4(50) | 3(12.5) | 0 | 0 | 2(50) | |
| | R | 54(76.1) | 8(47.1) | 26(81.3) | 4(50) | 21(87.5) | 2(100) | 3(100) | 2(50) | |
| AMK | S | 64(90.1) | 15(94.1) | 29(90.6) | 7(87.5) | 18(75) | 1(50) | 2(66.7) | 3(75) | |
| | R | 7(9.9) | 1(5.9) | 3(9.4) | 1(12.5) | 6(25) | 1(50) | 1(33.3) | 1(25) | |
| AMC | S | 3(4.2) | 0 | - | - | 0 | 0 | - | - | |
| | R | 68(95.8) | 17(100) | - | - | 24(100) | 2(100) | - | - | |
| TZP | S | 35(49.3) | 4(23.5) | 2(6.3) | 4(25) | 6(25) | 0 | 0 | 1(25) | |
| | R | 36(50.7) | 13(76.5) | 30(93.7) | 6(75) | 18(75) | 2(100) | 3(100) | 3(75) | |
| CZ | S | 2(2.9) | 0 | - | - | 0 | 0 | - | - | |
| | R | 69(97.1) | 17(100) | - | - | 24(100) | 2(100) | - | - | |
| CXM | S | 1(1.4) | 1(5.9) | - | - | 0 | 0 | - | - | |
| | R | 70(98.6) | 16(94.1) | - | - | 24(100) | 2(100) | - | - | |
| CRO | S | 3(4.2) | 2(11.8) | 0 | - | 0 | 0 | 0 | - | |
| | R | 68(95.8) | 15(88.2) | 32(100) | - | 24(100) | 2(100) | 3(100) | - | |
| CAZ | S | 4(5.6) | 2(11.8) | 0 | 1(12.5) | 0 | 0 | 0 | 0 | |
| | R | 67(94.4) | 15(88.2) | 32(100) | 7(87.5) | 24(100) | 2(100) | 3(100) | 4(100) | |
| CEP | S | 5(6) | 2(11.8) | 2(6.3) | 1(12.5) | 0 | 0 | 0 | 0 | |
| | R | 66(93) | 15(88.2) | 30(93.7) | 7(87.5) | 24(100) | 2(100) | 3(100) | 4(100) | |
| MER | S | 43(60.6) | 13(76.5) | 11(34.4) | 0 | 0 | 0 | 0 | | |
| | R | 28(39.4) | 4(23.5) | 21(65.6) | 8(100) | 24(100) | 2(100) | 3(100) | 4(100) | |
| CIP | S | 25(35.2) | 3(17.4) | 5(15.6) | 2(25) | 9(37.5) | 0 | 1(33.3) | 1(25) | |
| | R | 46(64.8) | 14(82.4) | 27(84.4) | 6(75) | 15(62.5) | 2(100) | 2(66.7) | 3(75) | |
| CHL | S | 41(57.7) | 12(70.6) | - | - | 7(29.2) | 2(100) | - | - | |
| | R | 30(42.3) | 5(29.4) | - | - | 17(70.8) | 0 | - | - | |
| SXT | S | 10(14.1) | 1(5.9) | 1(3.1) | - | 3(12.5) | 0 | 0 | - | |
| | R | 61(85.9) | 16(94.1) | 31(96.9) | - | 21(87.5) | 2(100) | 3(100) | - | |
| TE | S | 18(25.4) | 2(11.8) | 0 | - | 3(12.5) | 0 | 0 | - | |
| | R | 53(74.6) | 15(88.2) | 32(100) | - | 21(87.5) | 2(100) | 3(100) | - | |

Key: S = Sensitive, R = Resistance, AMP = Ampicillin; AMK = Amikacin; AMC = Amoxicillin/clavulanic acid; TZP = Piperacillin/tazobactam;

SXT = Sulphamethoxazole-trimethoprim; TE = Tetracycline; CIP = Ciprofloxacin; CHL = Chloramphenicol; GEN = Gentamycin; CRO = Ceftriaxone;

CAZ = Ceftazidime; CXM = cefuroxime; CZ = Cefazolin; CEP = Cefepime; MER = Meropenem

## Discussion

This study investigated MDR and carbapenemase-producing critical GNB in ICU environmental samples from two specialized hospitals in the Amhara region. The critical GNB is widely distributed on most of the environmental surfaces of ICUs, particularly in NICUs. This is an important way to evaluate the efficiency of surface cleaning and its impact on HAI. Simple measures, such as the correct sanitization of the environment and hand washing of healthcare personnel, have a significant impact on the reduction of neonatal morbidity and mortality, and the environment is a reservoir of microorganisms that can be transmitted to neonates via the hands and invasive devices [7].

In parallel with other studies [16–18], the present study found that most environmental surface types were contaminated with at least one MDR and CP critical GNB *(K. pneumoniae, E. coli, P. aeruginosa* and *Acinetobacter* species). This implies an extensive distribution of these pathogens in the ICUs at the study sites. This may be due to many factors: first, the low level of implementation of IPC practices in ICUs due to lack of regular staff training, active surveillance, and proper disinfectant supplies or reagents. Second, in the study setting, there was no rectal screening for MDR or CP GNB gut colonization on admission, every week, or at discharge [10]. Lastly, ICUs at the study settings have bay rooms or multiple bed rooms that make it difficult to apply most IPC measures such as contact precautions and the concept of the "patient's zone" developed by the WHO in its recommendations for hand hygiene. In ICUs with baby rooms or multiple-bed rooms, the patient's zone may be difficult to identify and adherence to hand hygiene may be lower [19].

In the present study, baby beds and incubators were highly colonized by MDR and CP critical GNB. One-third of neonatal deaths annually (680,000) are caused by infections, notably severe bacterial infections. AMR contributes to an estimated 140,000 neonatal deaths annually [3]. The high contamination rate recorded on the two aforementioned environmental surfaces of the NICUs may be due to some obvious reasons, as high numbers of neonates with different clinical conditions are frequently admitted for clinical attention and evaluation. This clinical practice requires the frequent presence and attention of breastfeeding and healthcare workers, thus increasing unit occupancy density, traffic and human activities. These issues worsen if NICUs health care institutions are teaching institutions [20]. In addition to the overcrowding, shared equipment with inadequate reprocessing, admission of up to three neonates in one baby bed or incubator, inadequate environmental cleaning, suboptimal hand hygiene compliance and quality of infrastructure may also play a role in the high level of contamination [20].

The present study showed that the most frequently isolated critical GNB was *K. pneumoniae* followed by *Acinetoacter* species, *E. coli* and *P. aeruginosa*. This result is similar to reports from India [21], Nigeria [22, 23] and Alexandria University [24], where *K. pneumoniae* was most commonly detected isolates from the environmental surfaces of the ICUs. In contrast to the present study, *Acinetobacter* species, *A. baumannni* was the most persistently detected isolate in ICUs [25, 26]. Moreover, a systematic review and meta-analysis that estimated the overall prevalence of bacterial contamination on inanimate surfaces and equipment in Ethiopia reported that *E. coli* was the most commonly detected isolate, followed by *P. aeruginosa* and *K. pneumoniae* [27]. This inconsistency with the results of other studies on the number and type of isolates may be due to differences in the number of people in the environment, amount of activity, amount of moisture, presence of material capable of supporting microbial growth, and type of environmental surface (for example, rough or smooth), and orientation in ICUs [11].

ICUs are the major sources of dissemination of antibiotic-resistant organisms, and selection pressure is highest for the emergence of resistance [28]. The nosocomial infection rate in ICUs

is two to five times higher than that in the general (non ICU) hospital population [10]. In this study, we observed a high resistance pattern to the commonly used antibiotics, ampicillin, amoxicillin/clavulanic-acid cotrimoxazole, tetracycline and cephalosporins. A similar pattern was reported at Tikur Anbessa Specialized Hospital, Ethiopia, which found that GNB was highly resistant to most of the tested antibiotics, such as ampicillin, ceftazidime, ceftriaxone, amoxicillin/clavulanic acid, and cefotaxime, ranging from 85 to 97.5% [29]. However, lower levels of resistance to amikacin were observed in this study. This variation in resistance to these antibiotics may be due to the availability of antibiotics in the pharmaceutics, differences in administration of intravenous and oral antibiotics, increased exposure to self-prescription, and frequent empirical prescription of these antibiotics by health professionals.

The most common multidrug resistant and principal carbapenemase producer reported in this study was *K. pneumoniae* with a rate of 71(89.9%) and 24(85.7%), respectively. In addition, the carbapenemase production rate was 50% in *P. aeruginosa* and *E. coli and* 14.3% in *Acinetobacter* species. This is marginally in line with a study conducted in Iran, in which the CP *K. pneumoniae* was 82.4%. [30]. Similarly, a study by Trivedi *et al.* reported that carbapenemase production was highest in *K. pneumoniae* followed by *E. coli*, *P. aeruginosa* and *Acinetobacter* species [21]. Isolation of drug-resistant *K. pneumoniae* from environmental surfaces in the NICU is an important risk factor for neonatal infections, including septicemia, urinary tract infections, pneumonia, and soft tissue infections. Neonatal sepsis is a primary cause of neonatal mortality within low- and middle-income countries (LMICs), with LMICs bearing the burden of 99% of global neonatal mortality [31]. A review of the characterization of antimicrobial-resistant GNB that cause neonatal sepsis in seven low- and middle-income countries showed that *K. pneumoniae* is an important cause of neonatal sepsis in LMICs and is predominantly found in Ethiopia, Nigeria, and Pakistan [32]. Moreover, some studies have shown that outbreaks of MDR and CP *K. pneumoniae* infections in NICUs have an environmental reservoir [33–36].

This study investigated environmental surfaces contaminated with carbapenemase-producing critical GNB isolates in the ICUs, assumed to be highly clean unit for giving care of critically ill patients. However, study used conventional biochemical tests rather than analytical profile index (API 20) or matrix-assisted laser desorption/ionization time-of-flight (MALDI-TOF) mass spectrometry because of unavailability and high expense, especially for non-fermenters. Molecular test for confirmation of carbapenemase-producing isolates was also not performed.

## Conclusions

Successful prevention of nosocomial infections requires investigation of the sources of environmental contamination and development of practical methods to prevent the spread of bacteria. The study performed at two institutions showed extensive distribution of MDR and CP critical GNB in ICUs. The high detection rates of MDR and CP *K. pneumoniae* on most environmental surfaces should be considered in control programs in the ICUs of the two hospitals. Urgent infection prevention and control action in the neonatal ICU is also important to prevent possible outbreaks.

## Acknowledgments

The authors would like to thank the University of Gondar Comprehensive Specialized Hospital, Felege Hiwot Comprehensive Specialized Hospital and the ICU healthcare workers. Many thanks to Solomon Taye (PhD) for editing and commenting on this manuscript.

## Author Contributions

**Conceptualization:** Mizan Kindu, Baye Gelaw.

**Data curation:** Mizan Kindu, Zemene Tigabu.

**Formal analysis:** Mizan Kindu.

**Investigation:** Mizan Kindu, Degu Ashagrie.

**Methodology:** Mizan Kindu, Degu Ashagrie.

**Resources:** Mizan Kindu.

**Software:** Mizan Kindu.

**Supervision:** Feleke Moges, Zemene Tigabu, Baye Gelaw.

**Writing – original draft:** Mizan Kindu.

**Writing – review & editing:** Baye Gelaw.

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
