## [Decision Letter · Decision Letter 0]

1 Aug 2023

PONE-D-23-20426Multidrug-resistant and carbapenemase-producing critical gram-negative bacteria isolated from the intensive care unit environment in Amhara Region, EthiopiaPLOS ONE

Dear Dr. Kindu,

Thank you for submitting your manuscript to PLOS ONE. After careful consideration, we feel that it has merit but does not fully meet PLOS ONE’s publication criteria as it currently stands. Therefore, we invite you to submit a revised version of the manuscript that addresses the points raised during the review process.

We look forward to receiving your revised manuscript.

Kind regards,

Arghya Das, MD

Academic Editor

PLOS ONE

Journal Requirements:

Additional Editor Comments:

Line 22: ‘amied’

Comment: typo

Lines 23-24: “……..gram negative bacteria (K. pneumoniae, E. coli, P. aeruginosa and Acinetobacter species) and their antibiotic resistance on ICU environmental surfaces”

Comment: Slightly modify the sentence “……..Gram negative bacteria (Klebsiella pneumoniae, Escherichia coli, Pseudomonas aeruginosa and Acinetobacter species) on ICU environmental surfaces and their antibiotic resistance……….”

Lines 46-47: “Moreover, the majority of carbapenemase-encoding genes are carried by mobile elements, such as plasmids, which will ease the vertical transmission of resistance across other GNB in the same setting”

Comment: In context of mobile genetic elements, probably the authors wanted to mean horizontal transmission of resistance determinants but wrote ‘vertical transmission’.

Line 73: “High-touch surfaces are those with frequent contact with the hands, which poses the greatest risk of transmission of microorganisms”

Comment: It will be better to mention that the authors targeted high-touch surfaces for environmental sampling before defining high-touch surfaces.

Line 74: ‘bed masters’

Comment: Probable typo

Line 82: “Single distinct colonies were isolated and purified by subculturing in fresh BAP medium to obtain pure culture isolates”

Comment: Authors should explain how they have chosen single distinct colonies for sub-culture. Environmental samples are likely to yield multiple organisms. Have authors considered culturing all types of isolates?

Lines 83-84: “Isolates were identified using different biochemical tests such as triple sugar iron agar, indole, motility, urease production, hydrogen sulfide production, citrate utilization, lysine decarboxylase tests, and oxidase strip tests”

Comment: Mentioned tests may not be enough to correctly identify Acinetobacter spp. Please explain.

Line 122: “A total of 384 environmental samples from neonates (246/384=64.1%) and other intensive care units (150/384=35.9%) were analyzed”.

Comment: Please check the statistics mentioned in the line.

Line 123: “Of 126(32.8%) culture positive samples, 162 critical GNB isolates……”

Comment: Write “From 126(32.8%) culture positive samples, 162 critical GNB isolates….”

Line 166: “Moreover, the magnitudes of MDR critical GNB were 75/95(78.9%) and 53/67(80.5%) at FHSCH, respectively”

Comment: Please check the above sentence. It seems incomplete.

Lines 166-168: “The neonatal intensive care unit of UoGCSH was more contaminated by MDR and CP-critical GNB than other ICUs. In contrast, at the FHCSH, the ICUs for pediatric, adult, and surgical critical care patients were more contaminated than the neonatal units”.

Comment: Just based on the proportions, it will not be prudent to mention on the level of contamination of different areas without doing relevant statistical analysis.

Lines 191-192: CP K. pneumoniae isolates were 25-87.5% resistant to amikacin, ciprofloxacin, chloramphenicol, piperacillin-tazobactam, gentamicin, sulfamethoxazole-trimethoprim and tetracycline, respectively.

Comment: It is difficult to follow the actual magnitude of resistance to different antibiotics from the above sentences. Authors are advised to either write to refer to Table 5 or mention resistance rates of all antibiotics individually.

Lines 283: “LMIC’

Comment: Please expand ‘LMIC’ where it is first used.

Table 2: ‘R’ refers to resistant to classes of antibiotics as per the footnote. But in reality, authors have considered types of antibiotics for calculation of ‘R1 to R15’ not antibiotic classes. Few statistics are little bit misleading because not all drugs were tested for all organisms. Therefore, the higher values of R like ‘R15’ should be denoted as ‘-‘ not as ‘zero’.

Table 3: Please delete the number of samples collected from the second row.

Table 4: The column totals of few columns are not matching with what mentioned in the table. Please check

Overall comment: Please mention the strengths and limitation of the present study toward the end of the Discussion.

Please also mention the measures which are planned to control environmental contamination with critical Gram-negative bacteria at the study centers.

Reviewers' comments:

Reviewer's Responses to Questions

**Comments to the Author**

1. Is the manuscript technically sound, and do the data support the conclusions?

Reviewer #1: Yes

Reviewer #2: Yes

2. Has the statistical analysis been performed appropriately and rigorously? 

Reviewer #1: Yes

Reviewer #2: No

3. Have the authors made all data underlying the findings in their manuscript fully available?

Reviewer #1: Yes

Reviewer #2: Yes

4. Is the manuscript presented in an intelligible fashion and written in standard English?

Reviewer #1: Yes

Reviewer #2: Yes

5. Review Comments to the Author

Reviewer #1: I would like to congratulate the authors for doing this valuable study. There are few clarifications needed which must be addressed in the manuscript. These are:

1. Please define MDR in the manuscript (resistance to how many antibiotics)

2. The samples have grown only gram negative organisms. No gram positive (S. aureus/ CONS) have been reported in any of the samples collected. This is in contrast to that seen in most other studies such as Darge, A., Kahsay, A.G., Hailekiros, H. et al. Bacterial contamination and antimicrobial susceptibility patterns of intensive care units medical equipment and inanimate surfaces at Ayder Comprehensive Specialized Hospital, Mekelle, Northern Ethiopia. BMC Res Notes 12, 621 (2019).

3. It would be more useful if in the discussion the authors could give a table of the observations of the other similar studies. This would help identify the local epidemiology of the pathogens based on the regions

4. It would be useful if the authors could add if any of the patients in the ICU developed infection with these organisms. A detailed profile of the infective pathogens and the infections caused in the ICU patients during this timeframe would help understand the relevance of these isolates.

Reviewer #2: The manuscript reports important results on the burden of Multidrug-resistant and carbapenemase-producing critical gram-negative bacteria isolated in hospital setups, particularly in ICU’s. Overall, the manuscript is well written and the data flow is clear. But some minor points need to be considered:

1. The terminology 'healthcare-associated infection' instead of 'hospital acquired infection' would be better suited here. Please make similar changes wherever applicable throughout the manuscript.

2. Sampling technique: Did the number of environmental sample collection took place at random, or any criteria (like sample size calculation) for sample collection was used? Please clarify in the main manuscript.

3. The acronym "BAP" appears in line no. 82, I imagine that the authors refer to: "blood agar plates." elaborate the acronym "BAP” since I did not find the acronym in another part of the text.

4. In antibiotic susceptibility result by disk diffusion method the authors have not mentioned anything about the intermediate results in the isolates. Were they counted as resistant or susceptible?

5. sCIM method for testing carbapenemase production in A. baumannii is not recommended by standard guidelines. Hence, few isolates should also be validated for the same by some other methods.

6. There are spacing errors in the text throughout the manuscript that should be corrected.

6. PLOS authors have the option to publish the peer review history of their article (what does this mean?). If published, this will include your full peer review and any attached files.

Reviewer #1: **Yes: **Dr Priyam Batra

Reviewer #2: No

---

## [Author Response · Author response to Decision Letter 0]

13 Sep 2023

Response to reviewers 

Response to the academic editor

Response: all the formatting for headings, tables, reference brackets in the body, etc… are corrected according to the journal style and all highlighted in the manuscript

Response: Edited and attached as suggested 

The manuscript is edited by Solomon Taye Sima, PhD; Postdoctoral Research Scientist; Center for Immunity and Inflammation; New Jersey Medical School; Rutgers University.

Response: All relevant data are within the manuscript.

A pre-existing iD in Editorial Manager is authenticated 

Response: corrected as suggested and highlighted in lines 118-121

Additional Editor Comments:

Response to additional editor comments 

Line 22: ‘amied’

Comment: typo

Response: corrected and highlighted in line 22

Lines 23-24: “……..gram negative bacteria (K. pneumoniae, E. coli, P. aeruginosa and Acinetobacter species) and their antibiotic resistance on ICU environmental surfaces”

Comment: Slightly modify the sentence “……..Gram negative bacteria (Klebsiella pneumoniae, Escherichia coli, Pseudomonas aeruginosa and Acinetobacter species) on ICU environmental surfaces and their antibiotic resistance……….”

Response: corrected as suggested and indicated in lines 23-24

Lines 46-47: “Moreover, the majority of carbapenemase-encoding genes are carried by mobile elements, such as plasmids, which will ease the vertical transmission of resistance across other GNB in the same setting”

Comment: In context of mobile genetic elements, probably the authors wanted to mean horizontal transmission of resistance determinants but wrote ‘vertical transmission’.

Response: corrected and highlighted in line 47

Line 73: “High-touch surfaces are those with frequent contact with the hands, which poses the greatest risk of transmission of microorganisms”

Comment: It will be better to mention that the authors targeted high-touch surfaces for environmental sampling before defining high-touch surfaces.

Response: corrected as suggested

Line 74: ‘bed masters’

Comment: Probable typo

Response: corrected and highlighted in line 73

Line 82: “Single distinct colonies were isolated and purified by subculturing in fresh BAP medium to obtain pure culture isolates”

Comment: Authors should explain how they have chosen single distinct colonies for sub-culture. Environmental samples are likely to yield multiple organisms. Have authors considered culturing all types of isolates?

Response: Thank you for the comment. Single distinct colonies means these colonies preliminary differentiated based on morphology on media. Morphologically deferent means that difference in size, shape, edge, lactose fermentation, mucoid or dry etc… Then, biochemical test was done for all morphologically deferent isolates. 

Lines 83-84: “Isolates were identified using different biochemical tests such as triple sugar iron agar, indole, motility, urease production, hydrogen sulfide production, citrate utilization, lysine decarboxylase tests, and oxidase strip tests”

Comment: Mentioned tests may not be enough to correctly identify Acinetobacter spp. Please explain.

Response: I really appreciate the comment. But, through the editing process, an additional test, the catalase test, was deleted from the list of mentioned tests.

In this study, most of the Acinetobacter colonies on MacConkey agar were non-lactose fermenters, with raised and smooth-edged colonies, but very few were sticky and flat. In addition to colony morphology characteristics, the above-mentioned biochemical tests, catalase and oxidase strip tests, were used to differentiate Acinetobacter spp.

Line 122: “A total of 384 environmental samples from neonates (246/384=64.1%) and other intensive care units (150/384=35.9%) were analyzed”.

Comment: Please check the statistics mentioned in the line.

Response: I accept the commented and highlighted in line 134

Line 123: “Of 126(32.8%) culture positive samples, 162 critical GNB isolates……”

Comment: Write “From 126(32.8%) culture positive samples, 162 critical GNB isolates….”

Response: corrected as suggested and highlighted in lines 135-136

Line 166: “Moreover, the magnitudes of MDR critical GNB were 75/95(78.9%) and 53/67(80.5%) at FHSCH, respectively”

Comment: Please check the above sentence. It seems incomplete.

Response: corrected and highlighted in line 165

Lines 166-168: “The neonatal intensive care unit of UoGCSH was more contaminated by MDR and CP-critical GNB than other ICUs. In contrast, at the FHCSH, the ICUs for pediatric, adult, and surgical critical care patients were more contaminated than the neonatal units”.

Comment: Just based on the proportions, it will not be prudent to mention on the level of contamination of different areas without doing relevant statistical analysis.

Response: I really accept the comment. But, when we did the analysis, the contamination rate among units was not statistically significant. Therefore, I corrected the statement and indicated in lines 165-168

Lines 191-192: CP K. pneumoniae isolates were 25-87.5% resistant to amikacin, ciprofloxacin, chloramphenicol, piperacillin-tazobactam, gentamicin, sulfamethoxazole-trimethoprim and tetracycline, respectively.

Comment: It is difficult to follow the actual magnitude of resistance to different antibiotics from the above sentences. Authors are advised to either write to refer to Table 5 or mention resistance rates of all antibiotics individually.

Response: I accept the comment and the corrected sentence highlighted in lines 186-187 and 189-190

Lines 283: “LMIC’

Comment: Please expand ‘LMIC’ where it is first used.

Response: corrected and highlighted in line 257

Table 2: ‘R’ refers to resistant to classes of antibiotics as per the footnote. But in reality, authors have considered types of antibiotics for calculation of ‘R1 to R15’ not antibiotic classes. Few statistics are little bit misleading because not all drugs were tested for all organisms. Therefore, the higher values of R like ‘R15’ should be denoted as ‘-‘ not as ‘zero’.

Response: Thank you for valuable comment. The table is amended as it was mentioned in the footnote and highlighted in the table

Table 3: Please delete the number of samples collected from the second row.

Response: corrected as suggested 

Table 4: The column totals of few columns are not matching with what mentioned in the table. Please check

Response: corrected and highlighted in the table

Overall comment: Please mention the strengths and limitation of the present study toward the end of the Discussion.

Response: corrected and highlighted in lines 262-266

Please also mention the measures which are planned to control environmental contamination with critical Gram-negative bacteria at the study centers.

Response: Thank you for the comment. As researcher, we already informed concerned bodies (head of hospitals) of two sites to take actions. 

Review Comments to the Author

Reviewer #1: 

1. Please define MDR in the manuscript (resistance to how many antibiotics)

Response: I accept the comment. It is added in manuscript and highlighted in line 93

2. The samples have grown only gram negative organisms. No gram positive (S. aureus/ CONS) have been reported in any of the samples collected. This is in contrast to that seen in most other studies such as Darge, A., Kahsay, A.G., Hailekiros, H. et al. Bacterial contamination and antimicrobial susceptibility patterns of intensive care units medical equipment and inanimate surfaces at Ayder Comprehensive Specialized Hospital, Mekelle, Northern Ethiopia. BMC Res Notes 12, 621 (2019).

Response: I really appreciate the comment given by the reviewer; however, my objective was only on the critical GNB as mentioned in the manuscript.

3. It would be more useful if in the discussion the authors could give a table of the observations of the other similar studies. This would help identify the local epidemiology of the pathogens based on the regions

Response: Thank you for the comment. But, as per my search similar studies have been already discussed. 

4. It would be useful if the authors could add if any of the patients in the ICU developed infection with these organisms. A detailed profile of the infective pathogens and the infections caused in the ICU patients during this timeframe would help understand the relevance of these isolates.

Response: Thank you for valuable comment. However, this paper is one part of my phd works. The final objective of this work was to see the transmission link or to detect the sources of infection for carbapenemase producing critical GNB in ICUs. Therefore, I collected environmental, clinical and rectal swab samples from ICUs. However, due to resource limitation for gene detection and whole genome sequencing other works were not yet finished. 

Reviewer #2: 

1. The terminology 'healthcare-associated infection' instead of 'hospital acquired infection' would be better suited here. Please make similar changes wherever applicable throughout the manuscript.

Response: corrected as suggested throughout the manuscript

2. Sampling technique: Did the number of environmental sample collection took place at random, or any criteria (like sample size calculation) for sample collection was used? Please clarify in the main manuscript.

Response: I accept the comment. It is clarified and highlighted in line 71

3. The acronym "BAP" appears in line no. 82, I imagine that the authors refer to: "blood agar plates." elaborate the acronym "BAP” since I did not find the acronym in another part of the text.

Response: I accept the comment. It is elaborated in line 80

4. In antibiotic susceptibility result by disk diffusion method the authors have not mentioned anything about the intermediate results in the isolates. Were they counted as resistant or susceptible?

Response: I accept the comment. Critical GNB isolates with intermediate results detected were very few in numbers, however, for analysis counted as resistant.

5. sCIM method for testing carbapenemase production in A. baumannii is not recommended by standard guidelines. Hence, few isolates should also be validated for the same by some other methods.

Response: I really appreciate comment given by the reviewer. However, due to lack of resource confirmatory molecular test was not done and this is indicated as limitation of this study 

6. There are spacing errors in the text throughout the manuscript that should be corrected.

Response: corrected as suggested

---

## [Decision Letter · Decision Letter 1]

27 Oct 2023

PONE-D-23-20426R1Multidrug-resistant and carbapenemase-producing critical gram-negative bacteria isolated from the intensive care unit environment in Amhara Region, EthiopiaPLOS ONE

Dear Dr. Kindu,

Thank you for submitting your manuscript to PLOS ONE. After careful consideration, we feel that it has merit but does not fully meet PLOS ONE’s publication criteria as it currently stands. Therefore, we invite you to submit a revised version of the manuscript that addresses the points raised during the review process.

We look forward to receiving your revised manuscript.

Kind regards,

Ali Amanati

Academic Editor

PLOS ONE

Journal Requirements:

Additional Editor Comments:

Editorial comments

Dear authors

The manuscript's overall presentation improved after amendments and is now ‎more readable‎. Although ‎your replies to the reviewers' concerns are satisfactory, ‎the manuscript ‎still needs to be improved.‎

1. Address reviewers' comment one-by-one.

Editor comments:

#Authors do not explicitly mention the specific study design used in the research.

#The following reviewers' comment has not been addressed correctly.

Line 73: “High-touch surfaces are those with frequent contact with the hands, which

poses the greatest risk of transmission of microorganisms”

Comment: It will be better to mention that the authors targeted high-touch surfaces for

environmental sampling before defining high-touch surfaces.

#Line 120:

The "Registration number" should be accessible by a valid link.

#Although the information presented in Table 2 is interesting, such a classification is not familiar in the field of clinical microbiology, so I recommend that you present it based on standard interpretations instead of such an explanation:

pan-sensitive, MDRو XDR, and PDR

Reviewers' comments:

Reviewer's Responses to Questions

**Comments to the Author**

1. If the authors have adequately addressed your comments raised in a previous round of review and you feel that this manuscript is now acceptable for publication, you may indicate that here to bypass the “Comments to the Author” section, enter your conflict of interest statement in the “Confidential to Editor” section, and submit your "Accept" recommendation.

Reviewer #1: All comments have been addressed

Reviewer #2: (No Response)

2. Is the manuscript technically sound, and do the data support the conclusions?

Reviewer #1: Yes

Reviewer #2: Yes

3. Has the statistical analysis been performed appropriately and rigorously? 

Reviewer #1: N/A

Reviewer #2: No

4. Have the authors made all data underlying the findings in their manuscript fully available?

Reviewer #1: Yes

Reviewer #2: Yes

5. Is the manuscript presented in an intelligible fashion and written in standard English?

Reviewer #1: Yes

Reviewer #2: No

6. Review Comments to the Author

Reviewer #1: I appreciate the work done by the team and recommend acceptance of the manuscript. I recommend that the authors should publish the clinical data as well

Reviewer #2: Most of the issues raised were successfully addressed by the authors. A few still need to be further clarified:

Line 18. Insert (HAI) after healthcare-associated infections and then the abbreviation can be used throughout the abstract.

Line 66. Owing to innovation or renovation?

Line 71. Better to use, samples from the hospital environment instead of environmental samples. Please do similar corrections wherever applicable

Line 92. I think second generation cephalosporins are also comes under extended-spectrum cephalosporins if so then, either write first & second generation cephalosporins or remove ‘non-extended cephalosporins’ or cefuroxime might be added to the extended-spectrum cephalosporins category in the above line 89-90.

Line 136. ‘162 critical GNB isolates were identified and 36 samples showed mixed growth. Add the percentage for these two results also

Line 152-153. ‘The most common MDR isolate was K. pneumoniae’. Remove this line, as the next statement is enough to explain that the K. pneumoniae was the most common MDR organism.

Line 177-178. I think it would be better to write, two different CP critical GNB rather than two types of CP critical GNB as its confusing.

Figure is of low resolution. Please endeavour to get a high quality figure which again can be changed in size without losing quality/resolution.

-Some minor grammatical as well as typographical errors are still present in the manuscript that needs to be corrected. I’m pointing a few as follows-

Line 18. Insert ‘are’ between ‘infections’ and ‘common’

Line 31. Change to ‘were detected on most of the surfaces from the hospital environment’

Line 32. Insert ‘from the’ between ‘especially’ and ‘baby bed sets and incubators’.

Line 34. Correct carpapenemase

Line 47. Replace ‘will’ with ‘might’ ease the horizontal transmission of

Line 80. use ‘into’ instead of ‘in’ before subculturing

Line 99 & 104. Correct 0.5McFarl to 0.5 McFarland

Line 121. Insert ‘of the’ between ‘environmental surface’ ‘ICUs’.

Line 151-152. Remove ‘of’ after 128(79 %)

7. PLOS authors have the option to publish the peer review history of their article (what does this mean?). If published, this will include your full peer review and any attached files.

Reviewer #1: **Yes: **Dr Priyam Batra

Reviewer #2: **Yes: **Dr. Swati Sharma

---

## [Author Response · Author response to Decision Letter 1]

13 Nov 2023

Response to reviewers

Author response: It is complete and corrected in the document

Author response to academic editor 

Editor comments: #Authors do not explicitly mention the specific study design used in the research.

Author response: Thank you for the comment. It is corrected and included in line 72.

Editor comment:-# The following reviewers' comment has not been addressed correctly. Line 73: “High-touch surfaces are those with frequent contact with the hands, which poses the greatest risk of transmission of microorganisms”. Comment: It will be better to mention that the authors targeted high-touch surfaces for

environmental sampling before defining high-touch surfaces.

Author response: Thank you for the comment. I mention the targeted high-touch surfaces in the study before the definition. Line 72-75 of the document.

Editor comment:#Line 120:The "Registration number" should be accessible by a valid link.

Author response: In our institution there is no online access for registration number to the ethical letter. The institution simply gives the hardcopy letter for the researchers. This document is already attached to PLOS ONE during submission.

Editor comment: #Although the information presented in Table 2 is interesting, such a classification is not familiar in the field of clinical microbiology, so I recommend that you present it based on standard interpretations instead of such an explanation:

pan-sensitive, MDRو XDR, and PDR

Authors Response:Thank you for the valuable comment. The focus of the current study is on MDR and carbapenemase producing isolates (already included in Table 2). On the other hand, according to the international experts to say XDR (nonsusceptibility to at least one agent in all but two or fewer antimicrobial categories i.e., bacterial isolates remain susceptible to only one or two categories) and PDR (nonsusceptibility to all agents in all antimicrobial categories) bacterial isolates should be tested against all or nearly all of the antimicrobial agents within the antimicrobial categories.

(https://www.sciencedirect.com/science/article/pii/S1198743X14616323)

The current study was not able to say PDR since the isolates were not tested to all of the antimicrobialagents within the antimicrobial categories listed by international experts such as aztreonam, fosfomycin and colistin.

Moreover, the table 2 was prepared by looking recent published papers in PLOS ONE journal (https://doi.org/10.1371/journal.pone.0264818 and https://doi.org/10.1371/journal.pone.0256556.t002), used R0, R1….

Finally, if I have to remove the columns for R0-R11, I am willing to do so.

Author response to the reviewers

Review Comments to the Author

Reviewer comment#1: I appreciate the work done by the team and recommend acceptance of the manuscript. I recommend that the authors should publish the clinical data as well

Author response: Thank you for all the valuable comments. I will send clinical data manuscript for publication the as soon as possible.

Reviewer comment #2: Most of the issues raised were successfully addressed by the authors. A few still need to be further clarified:

Line 18. Insert (HAI) after healthcare-associated infections and then the abbreviation can be used throughout the abstract.

Author response: Thank you for the comment. It is corrected as suggested and indicated in line 18,19 and 20.

Reviewer comment: Line 66. Owing to innovation or renovation?

Author response: It is renovation and corrected in line 66.

Reviewer comment: Line 71. Better to use, samples from the hospital environment instead of environmental samples. Please do similar corrections wherever applicable

Author response: It is corrected as suggested throughout the document.

Reviewer comment Line 92. I think second generation cephalosporins are also comes under extended-spectrum cephalosporins if so then, either write first & second generation cephalosporins or remove ‘non-extended cephalosporins’ or cefuroxime might be added to the extended-spectrum cephalosporins category in the above line 89-90

Author response: It is corrected by removing non-extended and extended cephalosporin replacing with first and second, and third and fourth generation cephalosporins, respectively. It is highlighted in line 90, 91, 93 and in table-1 in the main document.

Reviewer comment: Line 136. ‘162 critical GNB isolates were identified and 36 samples showed mixed growth. Add the percentage for these two results also

Authors response: corrected as suggested and included in line138

Reviewer comment: Line 152-153. ‘The most common MDR isolate was K. pneumoniae’. Remove this line, as the next statement is enough to explain that the K. pneumoniae was the most common MDR organism.

Author response: corrected as suggested

Reviewer comment: Line 177-178. I think it would be better to write, two different CP critical GNB rather than two types of CP critical GNB as its confusing

Authors response: corrected as suggested (line 178)

Reviewer comment:-Figure is of low resolution. Please endeavour to get a high quality figure which again can be changed in size without losing quality/resolution.

Author response: corrected as suggested

Reviewer comment:-Some minor grammatical as well as typographical errors are still present in the manuscript that needs to be corrected. I’m pointing a few as follows-

Line 18. Insert ‘are’ between ‘infections’ and ‘common’

Authors response: corrected as suggested in line 18

Reviewer comment:-Line 31. Change to ‘were detected on most of the surfaces from the hospital environment’

Authors response: corrected as suggested in line 31 and 32

Reviewer comment:-Line 32. Insert ‘from the’ between ‘especially’ and ‘baby bed sets and incubators’.

Authors response: corrected as suggested in line 32

Reviewer comment:-Line 34. Correct carpapenemase

Author response: corrected as suggested in line 34

Reviewer comment Line 47. Replace ‘will’ with ‘might’ ease the horizontal transmission of

Author response: corrected as suggested in line 47

Reviewer comment: Line 80. use ‘into’ instead of ‘in’ before subculturing

Author response: corrected as suggested in line 81

Reviewer comment:-Line 99 & 104. Correct 0.5McFarl to 0.5 McFarland

Authors response: corrected as suggested in line 101 and 106

Reviewer comment:-Line 121. Insert ‘of the’ between ‘environmental surface’ ‘ICUs’.

Authors response: corrected as suggested in line 123

Reviewer comment: Line 151-152. Remove ‘of’ after 128(79 %)

Author response: corrected as suggested

---

## [Editor Report · Decision Letter 2]

20 Nov 2023

Multidrug-resistant and carbapenemase-producing critical gram-negative bacteria isolated from the intensive care unit environment in Amhara Region, Ethiopia

PONE-D-23-20426R2

Dear Dr. Mizan Kindu,

We’re pleased to inform you that your manuscript has been judged scientifically suitable for publication and will be formally accepted for publication once it meets all outstanding technical requirements.

Kind regards,

Ali Amanati

Academic Editor

PLOS ONE

Additional Editor Comments (optional):

The authors have satisfactorily addressed my concerns.‎ I thank the authors for ‎their very detailed ‎‎replies to my comments.‎

---

## [Editor Report · Acceptance letter]

23 Nov 2023

PONE-D-23-20426R2 

Multidrug-resistant and carbapenemase-producing critical gram-negative bacteria isolated from the intensive care unit environment in Amhara Region, Ethiopia 

Dear Dr. Kindu:

I'm pleased to inform you that your manuscript has been deemed suitable for publication in PLOS ONE. Congratulations! Your manuscript is now with our production department. 

Kind regards, 

on behalf of

Professor Ali Amanati 

Academic Editor

PLOS ONE